# A Program of Life-Style Modification Improved the Body Weight and Micronutrient Status in Obese Patients after Bariatric Surgery

**DOI:** 10.3390/nu15173807

**Published:** 2023-08-30

**Authors:** Marta Crespo-Yanguas, Jairo Lumpuy-Castillo, Cristina Espadas, Carmen Aragón-Valera, Clotilde Vázquez, Óscar Lorenzo

**Affiliations:** 1Division of Endocrinology, Hospital Fundación Jiménez Díaz, 28040 Madrid, Spain; marta.crespo@fjd.es (M.C.-Y.); caragonva@quironsalud.es (C.A.-V.); clotilde.vazquez@quironsalud.es (C.V.); 2Laboratory of Diabetes and Vascular Pathology, IIS-Fundación Jiménez Díaz, Universidad Autónoma, 28040 Madrid, Spain; jairo.lumpuy@estudiante.uam.es (J.L.-C.); cristina.espadas@quironsalud.es (C.E.); 3Biomedical Research Network on Diabetes and Associated Metabolic Disorders (CIBERDEM), Carlos III National Health Institute, 28029 Madrid, Spain

**Keywords:** program of lifestyle modification, bariatric surgery, obesity, micronutrients

## Abstract

Introduction: Bariatric surgery is an efficient approach to rapidly reduce morbid obesity and associated comorbidities. However, approximately one-fourth of patients experience weight and comorbidity recurrence, and both obesity and bariatric surgery can lead to micronutrient deficiencies. Implementing a structured program of lifestyle modification (PLM) might enhance weight loss and improve micronutrient status. Methodology: A total of 121 severely obese patients underwent Roux-en-Y gastric bypass (RYGB). Among them, 71 adhered to a PLM involving dietary changes (low- and very-low-calorie Mediterranean diets) and physical exercises (aerobic and resistance training) both before and after surgery, while 50 patients followed a conventional protocol. Anthropometric measurements and serological parameter quantifications were conducted throughout the procedures. Results: The obese study population, primarily female (76.9%), with an average age of 47.11 ± 9.68, and a body mass index (BMI) of 44.68 ± 5.08 kg/m^2^, underwent either RYGB with a PLM or a conventional procedure. Before surgery, the PLM group exhibited significant reductions in body weight (6.3%) and phosphoremia compared to the conventional protocol (0.78%). Post-RYGB, the PLM group demonstrated shortened in-hospital stays and further BMI reductions (−16.12 kg/m^2^) that persisted for up to 2 years. Furthermore, the PLM group experienced increased plasma vitamin D levels (14.79 ng/mL vs. 1.2 ng/mL) for up to 2 years, as well as elevated folic acid (1.52 vs. −0.29 ng/mL) and phosphorus (0.48 vs. 0.06 mg/dL) levels at 1 month and 1 year after intervention, respectively. Notably, these effects were independent of weight loss. Conclusions: Initiating a structured PLM from the early stages of patients’ preparation for RYGB could enhance and extend the benefits of weight loss and positively impact micronutrient (vitamin D, phosphorus, and folic acid) status in obese patients.

## 1. Introduction

By 2030, more than half of the adult population will suffer from excessive body weight [1]. In particular, severe class-III obesity, typically when the body mass index (BMI) reaches over 40 kg/m^2^, has increased in the last few decades [2]. Morbid obesity is an important risk factor for the development of metabolic and cardiovascular diseases, reducing the life expectancy of patients by 6.5 to 13.7 years [3]. In fact, there is a J-shaped relationship between BMI and all-cause mortality in both men and women, with the highest hazard reached when BMI ≥ 40.0 kg/m^2^ [2]. Common comorbidities of obesity include hypertension (HTN), high cholesterol, type 2 diabetes (T2DM), and obstructive sleep apnoea/hypopnea syndrome (OSAHS), as well as heart disease, venous stasis disease, non-alcoholic fatty liver disease, arthritis, and psychosocial stress [4]. Furthermore, obesity is linked to a deficiency of macronutrients (i.e., proteins) and primarily micronutrients such as minerals, vitamins, and related hormones (i.e., leptin, insulin, and ghrelin) that play pivotal roles in obesity-associated comorbidities. Specially, vitamin D insufficiency followed by a deficit in folic acid (vitamin B9) may represent the most prevalent micronutrient disorder in morbidly obese patients. These alterations impact phosphorus–calcium metabolism and parathyroid hormone (PTH) secretion [5]. Importantly, changes in mineral metabolism have been linked to metabolic and cardiovascular pathologies. These modifications can have lethal implications for individuals with obesity [6,7].

However, the multifactorial aetiology of obesity, where genetic and hormone alterations may converge with environmental factors (i.e., eating disorders, physical inactivity, psychological stress, and circadian deregulations), lessens the efficiency of potential treatments. In this regard, bariatric surgery is the main strategy to attenuate severe and complex obesity. Current approaches can be highly efficient and safe, inducing a substantial long-term weight loss with minimal perioperative mortality (less than 0.1%) [8]. Additionally, the reduction in weight can be associated with improvements in important comorbidities such as T2DM, fatty liver disease, dyslipidaemia, HTN, and OSAHS [9]. The underlying mechanisms include changes in gastrointestinal anatomy and motility, modifications in diet and behaviour, gut hormones, bile acid flow, and gut bacteria [10]. Specially, the Roux-en-Y gastric bypass (RYGB) can achieve significant weight loss through both gastric restriction and nutrient malabsorption. This intervention creates a small gastric reservoir connected to the small intestine by means of a “Y de Roux” loop. The expected percentage of weight loss after 2 years of RYGB is 74–80% [11]. However, a weight regain (up to 30% of maximum loss) and a post-surgical decrease in micronutrient absorption has been described, even after mineral and vitamin supplementation [12,13]. Thus, those patients who do not reach the desirable body weight after the RYGB or those who exhibit micronutrient deficiencies at any pre-surgical or post-surgical stage could require additional interventions. In this sense, a program of lifestyle modification managed by bariatric nutritionists and physiotherapists from pre- to post-operative phases could help attenuate these deficiencies while further increasing and prolonging the weight loss achieved by the RYGB. Lifestyle modifications can incorporate changes in diet and physical activity toward a healthier diet and performance of specific aerobic and resistance exercises [14]. A combination of both strategies could potentially yield higher benefits for obese patients undergoing bariatric surgery than either can yield by themselves. In addition, a nutritional follow up may avoid metabolic imbalances related to the rapid weight loss, such as reduction in fat-free mass, malnutrition, and micronutrient deficiencies [15].

## 2. Methodology

### 2.1. Bariatric Surgery for Severe Obesity

This interventional prospective study focused on a cohort of 121 patients with severe obesity who underwent bariatric surgery using the RYGB technique at Fundación Jiménez Díaz Hospital. Patients followed a multidisciplinary intervention involving departments of Endocrinology and Nutrition, Psychiatry, Pneumology, Rehabilitation, Surgery, and Gastrointestinal, in accordance with international guidelines [16]. Two cohorts of patients were analysed: one before the establishment of a Program of Lifestyle Modification (PLM) (*n* = 50), and another subsequent cohort who followed this programme (*n* = 71). Our main aim was to evaluate whether the inclusion of a structured PLM could lead to improvements in BMI during the pre-surgical stage and the weight loss achieved with the RYGB. Additionally, we wondered whether the micronutrient status could be preserved during peri-surgical phases. The sample size was calculated using GRANMO software (v7.2, Barcelona, Spain) and taking into account an alpha risk of 0.05 (two-sided test), a beta risk of 0.2, a standard deviation of 2.5, a minimum expected difference of 1.5, and a dropout rate of 0.1, in accordance with a previous meta-analysis of anti-obesity interventions [17]. A minimum of 49 patients were recommended for each group. We included 50 and 71 individuals in the control and PLM groups, respectively.

#### 2.1.1. Selection of Patients

During the first clinical visit (V0) conducted by endocrinologists, patients with severe obesity were selected for a RYGB program (Figure 1). The inclusion criteria were an age of 18–65 years, a BMI ≥ 40 kg/m^2^ or BMI ≥ 35 kg/m^2^ with high-risk comorbidities (i.e., uncontrolled T2DM, HTN, dyslipidaemia and/or OSAHS, or physical incapacity), having maintained obesity for 5 years, and an ability to attend and follow the peri-surgical recommendations. Exclusion criteria included patients with serious psychiatric illness, endocrinological diseases responsible for obesity, cholelithiasis, hepatic steatosis, neoplastic malignancies, or medical contraindications. Once patients met requirements, they were evaluated by endocrinologists during three visits at standardized intervals (at the -8th, -4th, and -1st month before the RYGB) along the pre-surgical stage. They assessed the BMI progression, the related comorbidities and pharmacology, and plasma parameters (glucose and lipid profile, vitamins, and ions). They also considered the potential supplementation of vitamin D (calcifediol (0.26 mg) and cholecalciferol (800 UI)) and calcium (1 g), as advised by the international guidelines [16,18]. Other pre-surgical assessments were also scheduled. The pneumologists performed a baseline polysomnography for a potential OSAHS diagnosis, and psychiatrists assessed the presence of eating disorders that require psychiatric treatment prior to surgery. Lastly, the radiologists analysed possible cholelithiasis, hepatic steatosis, and neoplastic malignancies using abdominal ultrasound. They also assessed the presence of ulcers and *Helicobacter pylori* infection via gastrointestinal endoscopy. Subsequently, at the pre-surgical visit (V1) (30 days before surgery), the final anthropometric and plasma parameters, and any possible comorbidities and treatments, were recorded before undergoing the RYGB.

#### 2.1.2. The Roux-en-Y Gastric Bypass

Through the RYGB procedure, surgeons created a small gastric reservoir of 15–30 mL connected to the small intestine by means of a “Y de Roux” loop, with a biliary loop length of 70 cm and a food loop of 150 cm. Subsequently, the gastro-jejunal anastomosis was performed with a 25 mm circular stapler [19]. The RYGB was successfully achieved for all patients. However, potential complications associated with the intervention (within 30 days and from 30 days to 24 months after surgery) and the in-hospital stay were recorded. After the intervention, during the following five visits at the 1st, 3rd, 6th (V2), 12th (V3), 18th, and 24th (V4) month after surgery, endocrinologists evaluated patient outcomes including BMI, plasma parameters, and the resolution of comorbidities. Additionally, an extra visit with a psychiatrist was scheduled for the 3rd month after the RYGB. Consequently, anti-HTN treatments were maintained in 9 patients in both the control (18%) and PLM (12.7%) groups (*p* = 0.418), while anti-diabetic drugs were preserved in 3 (6.0%) and 1 (1.1%) subjects in the control and PLM groups, respectively (*p* = 0.305). Anti-hyperlipidaemics were retained in 6 control patients (12%) and in 2 PLM patients (2.8%) (*p* = 0.064). Moreover, lifelong supplementation with a multivitamin/multimineral formulation (1 pill/day (vitamins B, C, A, D, K, E, folic acid, biotin, Fe^++^, Mg^++^, Zn^++^, Se^++^, Mn^++^, Mo^++^, I^−^, Cu^+^, and Cr^++^)), vitamin D (calcifediol (0.26 mg)), and calcium (1 g) was prescribed for all patients, following nutritional guidelines [18]. Adherence to these treatments was confirmed during subsequent visits and sessions. All procedures adhered to the Guidelines for Perioperative Care in Bariatric Surgery: Enhanced Recovery After Surgery (ERAS) [20].

### 2.2. Anthropometric and Biochemical Parameters

The endocrinologists recorded the anthropometric data and obtained blood samples in all clinical visits, from V0 to V4. Information such as age, sex, clinical history, comorbidities, and pharmacological treatments was included. The body weight (up to 220 kg) was measured using the mechanical column scale with an eye-level beam (SECA-711; approval class III). A measuring telescopic rod with a range from 60 to 200 cm was used for assessing both height and weight in one step. This precision equipment was specially designed for weighing overweight patients (graduation: 100 g). A special coating reliably prevented the scale from fading. The unabridged declaration of conformity can be found at: www.seca.com. Nevertheless, the equipment was calibrated prior to data collection for each patient. Then, the BMI was estimated as a measure of body mass based on height and weight for adult men and women (kg/m^2^). Additionally, the Percentage of Total Weight Loss [%TWL: (initial weight − current weight)/(initial weight) × 100], Percentage of Excess Weight Loss [%EWL: (initial weight − current weight)/(initial weight − ideal weight) × 100], Percentage of Excess BMI Loss [%EBMIL: (initial BMI − current BMI)/(initial BMI − 25) × 100], and Percentage of Expected Excess BMI Loss [%EEBMIL: (initial BMI − current BMI)/(initial BMI − expected BMI) × 100, being Expected BMI: (0.33 × initial BMI + 14)] [21,22] were calculated. The circulating molecules were serum ions (sodium, potassium, phosphorus, iron, and calcium), vitamins (B12 and D, as 25(OH)D), parathyroid hormone (PTH), folic acid, glucose, glycosylated haemoglobin (HbA1ac), and the lipid profile (total cholesterol (TC), triglycerides (TG), low-density lipoprotein cholesterol (LDL-C), and high-density lipoprotein cholesterol (HDL-C)). The measurements were conducted at the Department of Clinical Analysis of the Fundación Jiménez Díaz Hospital. The date of surgery was also registered to determine whether it belonged to the sunlight season, from March 20th to September 23rd, or the non-sunlight season, from September 23rd to March 20th, in order to evaluate the influence of sun exposure on plasma biomarkers [23].

### 2.3. The Program for Lifestyle Modification

The PLM consisted of six/seven pre-surgical sessions with nutritionists and physiotherapists (4 in groups, and 2–3 individuals) and five individual post-surgical sessions, spanning approximately 12 months before and after the RYGB (Figure 1). In summary, during the first four sessions, the bariatric nutritionists explained the composition of healthy diets and nutrients, offering practical guidance for achieving balanced meals and healthy eating/drinking behaviours. Following the PREDIMED study [24], they recommended a hypocaloric Mediterranean diet (1500–2000 kcal/day) comprising fish, vegetables, fruits, legumes, olive oil, and whole grains. Protein content constituted 18–20% of total energy intake (TEI), carbohydrates accounted for 48% of TEI, and fat contributed 32–34% of TEI (with 10% from saturated fat). Moreover, personalized physical exercises were prescribed by physiotherapists for each individual. Aerobic activities (e.g., walking, biking, elliptical) were advised for at least 150 min/week, aiming for an intensity of 5–6 on the Borg scale [25]. Muscle strengthening exercises (e.g., using TheraBand loops) targeting upper and lower limbs (biceps, triceps, quadriceps, and hamstrings) were designated for 2–3 sessions/week. Adherence to recommended diet and physical activity was confirmed during subsequent individual sessions. Furthermore, in the last month of pre-surgical preparation (from V1 to surgery), nutritionists modified the hypocaloric diet to a very- low-calorie diet (600–800 kcal/day; proteins, 37% of TEI; carbohydrates, 43% of TEI; and fat, 20% of TEI (with 7.8% from saturated fat)) to reduce the size of the liver and facilitate the surgery [26].

Following the RYGB, bariatric nutritionists conducted an additional five individual sessions for the PLM patients. In the first session (occurring at the 1st month post-RYGB), personalized diets and physical exercise were commended. The post-surgery diet consisted of a series of consecutive hypocaloric diets with varying textures: an initial phase (15 days) featuring a liquid diet (600–800 kcal/day; proteins contributing 35% of TEI, carbohydrates contributing 53% of TEI, and fat contributing 12% of TEI (with 4.9% from saturated fat)), followed by an intermediate period (15 days) with a mashed diet (800–1000 kcal/day; proteins accounting for 32% of TEI, carbohydrates for 41% of TEI, and fat for 27% of TEI with 5.3% as saturated fat). Lastly, a final phase introduced a solid diet (1200 kcal/day; proteins making up 30% of TEI, carbohydrates comprising 38% of TEI, and fat contributing 32% of TEI (with 6.0%, saturated fat)), progressively adjusted to food tolerance. Recognizing that protein deficiency (serum albumin < 3.5 mg/dL) represents a significant macronutrient concern linked to the RYGB due to the acquired intolerance for protein-rich nutrients [10], the initial protein intake was established at 60–80 g/day. Subsequently, it was elevated to 90–120 g/day by incorporating additional protein-rich foods (such as eggs, meat, and lentils). Correspondingly, the initial recommended physical activity encompassed aerobic training (e.g., walking) for 30 min/day during the first 6–8 weeks post-RYGB, followed by the gradual inclusion of gentle strengthening exercises. In subsequent sessions (occurring at the 2nd, 4th, 9th, and 13th month post-RYGB), assessments were conducted to evaluate food tolerance and feeding frequency, while also confirming adherence to the prescribed diet and physical activity. The Ethical Committee of Clinical Studies of Fundación Jiménez Díaz Hospital granted approval for this study (Ref.: PIC170-18).

## 3. Data Analysis

Statistical comparisons between groups were conducted using appropriate tests based on the distribution of the variables. The normality of quantitative variables was assessed using the Kolmogorov–Smirnov test with Lilliefors correction. For variables demonstrating a normal distribution, mean values and standard deviation were calculated and compared utilizing the Student’s *t*-test for independent samples. Variables exhibiting a non-normal distribution were described with medians and interquartile ranges (IQR), and comparisons were carried out using the Mann–Whitney U test for independent samples. Categorical variables were presented through absolute frequencies and percentages and subjected to comparison through the Chi-squared test or Fisher exact test. Subsequently, relationships between variables were explored employing either simple and multiple linear regression or quantile regression, depending on variable distribution. Statistical significance was established as *p* < 0.05. All statistical analyses were performed utilizing the Statistical Package for the Social Sciences (SPSS v.26.0., IBM, Armonk, NY, USA).

## 4. Results

### 4.1. Characterization of the Morbid Obese Population

The overall population comprised 121 individuals with obesity, primarily female (76.9%) with a mean age of 47.11 years old, and a BMI of 44.68 ± 5.08 kg/m^2^. After undergoing clinical evaluation at V0, patients were directed toward either the conventional RYGB procedure (*n* = 50; control) or the PLM approach (*n* = 71) (Figure 1). At this stage, no significant differences were observed between the groups concerning age, sex, and BMI (Figure 2A), nor did differences manifest in most of the plasma parameters (Table 1). Only levels of creatinine and vitamin D exhibited elevation in control patients relative to those in the PLM group. Moreover, the prevalence of obesity-linked comorbidities (Figure 2B), utilization of pharmacological treatments (excluding metformin) (Appendix A), or engagement with continuous positive airway pressure (CPAP; not shown) appeared consistent across both groups. Also, the supplementation of vitamin D and calcium was comparable in both control and PLM groups (calcifediol usage in 53.7% and 48.4% of control and PLM individuals, respectively (*p* = 0.3), and cholecalciferol plus calcium administration in 2.4% and 4.7% of control and PLM patients, respectively (*p* = 0.39)). Then, while control participants received follow up from endocrinologists through conventional visits, PLM patients received supplementary guidance from nutritionists and physiotherapists to facilitate the adoption of healthy habits.

### 4.2. Lifestyle Modification at the Pre-Surgical Stage

On average, after a span of 13.6 months (14.34 ± 6.8 and 13.0 ± 6.0 months for control and PLM subjects, respectively), all patients were assessed during the pre-surgical visit (V1). In comparison to V0, the PLM group exhibited a greater extent of weight loss relative to the control group (−2.76 Kg/m^2^ vs. −0.35 Kg/m^2^, respectively) (Table 2). The %TWL and the %EWL also showed significant elevation in the PLM group (6.37% and 16.33% vs. 0.78% and 1.93%, respectively). At the plasma level, no differences between groups were observed for most of the biochemical parameters. However, PLM patients demonstrated a significant rise in phosphorus levels, as well as tendencies toward higher concentrations of vitamin D and folic acid, and lower levels of PTH (Table 2). Employing lineal regression for adjustments of age, sex, differential BMI (V1-V0), initial vitamin D treatment, and the date/season of surgery, phosphorus levels remained notably elevated in the PLM group (correlation coef. = 0.35, *p* = 0.04, 95% CI). Thus, the implementation of PLM appeared to facilitate increased weight loss and higher phosphoremia in individuals with morbid obesity during the pre-surgical stage.

### 4.3. Lifestyle Modification at the Post-Surgical Phase

All patients successfully underwent RYGB, although a few surgical complications were found. During the perioperative period, gastric perforation, dyspnoea, and digestive infections occurred in 2.5–5.8% of subjects. In the post-surgical phase, occurrences of cholecystectomy, hernia/eventration, and intestinal obstruction were observed in 2.5–9.9% of them (Figure 3A). However, no significant differences in these complications were discerned between the two groups. Notably, the in-hospital stay after RYGB was significantly shorter (≤4 days) for 65.9% of PLM patients compared to 34.1% of the control (*p* = 0.008; Figure 3B). Furthermore, a heightened tendency towards the complete resolution of OSAHS, TDM2, dyslipidaemia, and HTN was evident among PLM patients (Figure 4A). Hence, PLM might contribute to the reduction in the in-hospital stay for obese patients post-RYGB, potentially impacting their associated comorbidities. In addition, both BMI and micronutrient levels showed distinct changes in control and in PLM subjects following surgery.

#### 4.3.1. Reduction in the Body Mass Index after the RYGB and PLM

Notably, the reduction in body weight was significantly more pronounced 6 months following RYGB (V2) among PLM participants (BMI: −12.7 kg/m^2^) compared to the control group (BMI: −11.2 kg/m^2^) (Table 3). This divergence persisted 1 year after surgery (V3) and was sustained for up to 2 years (V4) post-intervention (−16.1 kg/m^2^ vs. −13.5 kg/m^2^, respectively). There were no significant differences in weight loss across visits (V4-V1), suggesting that the PLM’s impact was more prominent during pre-surgical stages. Similarly, the %TWL and the %EEBMIL were consistently higher in PLM subjects up to V3 (Table 3). Additionally, adjustments were made for age, sex, and the in-hospital stay in analysing the differences in progressive weight loss. Remarkedly, the reduction in BMI from V0 to V4 remained significantly more substantial in the PLM group compared to the control (Table 4). These findings suggest that the implementation of PLM may lead to further BMI reduction in patients undergoing RYGB, even up to 2 years post-surgery.

#### 4.3.2. Mineral Metabolism after the RYGB and PLM

Both obesity and bariatric surgery can contribute to deficiencies in vitamins and minerals [27]. However, our study revealed significantly higher levels of vitamin D in the PLM group (14.79 ng/mL) compared to the control (1.2 ng/mL) for up to 2 years post-RYGB (V4) (Table 5). Also, phosphorus and folic acid levels showed increases after PLM intervention for up to 1 year (V3; 0.48 vs. 0.03 mg/dL) and 6 months (V2; 1.52 vs. −0.29 ng/mL), respectively. Notably, even after adjusting for factors such as age, sex, differential BMI (V1 vs. V0), initial vitamin D supplementation, and the date/season of surgery, these differences roughly persisted (Table 6). Therefore, adherence to the PLM may have a direct positive impact on plasma vitamin D and phosphorus levels following RYGB surgery.

## 5. Discussion

The success of bariatric surgery, defined as a reduction in body weight and associated comorbidities while minimizing complications and maintaining metabolic homeostasis, relies not solely on surgery execution but also on the peri-surgical preparation of patients. An integrated PLM that encompasses guidance and continuous monitoring of low- and very-low-hypocaloric Mediterranean diets, along with a regime of physical exercise before and after bariatric surgery, may play a pivotal role in achieving greater body weight reduction and increased levels of essential micronutrients in these subjects.

Evidence demonstrates that a PLM may facilitate weight loss and the reduction in risk factors for metabolic and cardiovascular diseases, yet there is insufficient solid evidence regarding its effectiveness in the context of bariatric surgery. Currently, consensus guidelines do not incorporate perioperative conditioning for individuals undergoing bariatric procedures [28,29,30]. Implementing a PLM with a rigidly set duration, frequency, and intensity could potentially lead to surgery delays and cancellations, or inadvertently reinforce obesity and stigma in certain patients [31]. Moreover, the additional weight loss potentially brought about with a PLM is often modest and short-term in nature [32]. In this context, several studies have indicated a reverse correlation between pre-surgical stages (target of 5–10%) and surgery-associated mortality and complications [33,34]. In our study, patients who adhered to a structured PLM exhibited a higher pre-surgical weight loss (6.3%) compared to the control group (0.7%). This effect persisted for up to 24 months after the RYGB, regardless of their age, sex, and in-hospital stay. A similar outcome was reported by Giordano et al. [35] after 12 months post-surgery. Furthermore, our patients demonstrated a shorter post-surgical in-hospital stay and a greater likelihood of reducing complications (e.g., digestive infection and hernia) as well as resolving comorbidities such as T2DM, dyslipidaemia, and HTN. Therefore, obese patients who adhered to a PLM consisting of hypocaloric diets combined with aerobic/resistance training may contribute to pre-surgical BMI reduction, which could play a pivotal role in enhancing RYGB outcomes for up to 2 years post-intervention. Indeed, hypocaloric Mediterranean diets rich in minimally processed plant-based foods, abundant in monounsaturated fat from olive oil, and low in saturated fat and meats have shown potential to induce BMI reduction and improve cardiovascular health [36]. Additionally, a combination of aerobic and resistance exercises has been demonstrated to yield greater benefits for weight loss and cardiorespiratory fitness compared to each training approach alone [37]. In sum, integrated behavioural weight management programs that incorporate dietary and physical activity modifications in conjunction with bariatric surgery may offer heightened and enduring effectiveness in terms of weight loss and comorbidity management, surpassing the impact of individual diet or exercise interventions [38].

In addition, this PLM may be useful to maintain the micronutrient status in obese patients undergoing RYGB. Vitamin and mineral deficiencies have been widely described in obesity and after bariatric surgery, regardless of the type of procedure and despite exogenous supplementation [27]. However, the PLM significantly induced higher levels of plasma phosphorus already at the pre-surgical stage and elevated vitamin D, folic acid, and phosphorus in the postoperative phases. These improvements were independent of age, sex, changes in BMI, the initial supplementation of vitamin D, and the date/season of surgery. There was also a tendency to increase calcium while reducing PTH. Interestingly, the enhanced levels of vitamin D were preserved for up to 2 years after the RYGB. In this regard, the hypocaloric Mediterranean diet has been positively correlated with vitamin D levels, likely due to its anti-inflammatory and antioxidant properties [39]; but also, endurance exercises, not resistance exercises, could significantly increase vitamin D in patients deficient in vitamin D, likely due to the regulation of its metabolites [40]. Vitamin D is a fat-soluble vitamin essential for ion metabolism and the regulation of immune, cardiovascular, and endocrine responses [41]. The active isoform, 1,25(OH)2D, acts on phosphorus–calcium metabolism and interacts with the vitamin D receptor to exert non-calcaemic actions, such as the attenuation of the renin–angiotensin–aldosterone system and improvement in the lipid profile. However, in obesity, vitamin D becomes trapped within adipocytes, leading to its clearance from plasma. Each increased unit of BMI produces a 1.15% reduction in plasma vitamin D [42]. The lack of vitamin D also provokes elevated PTH levels, decreasing insulin sensitivity, initiating lipogenesis, and increasing fat mass. It also correlates with lower folic acid levels, which is an essential co-factor for DNA synthesis and repair [43]. Furthermore, after RYGB, the rapid weight loss and mechanical restriction and/or malabsorption of nutrients promote fat loss and further shortages of vitamin D and folic acid [44,45]. Thus, mineral and vitamin deficiencies contribute to producing negative effects on cardiovascular and metabolic diseases like obesity [46]. In this sense, a structured PLM from the early stages of conditioning to RYGB could not only enhance and prolong weight loss but also help maintain micronutrient levels to accelerate the resolution of comorbidities and metabolic recovery.

## 6. Limitation of the Study

Likely, a larger study would have yielded significant differences in data between a conventional protocol and the PLM regarding the resolution of comorbidities and the levels of other minerals and vitamins. As pre-surgical weight loss could play a critical role in achieving final improvements in body weight and micronutrient status, a comparison with a group of patients who adhered exclusively to a pre-surgical PLM might be necessary. Along these lines, mechanisms to ensure adherence to dietary and physical activity recommendations might also be included. Moreover, exploring other dietary approaches or modifications to feeding frequency (such as intermittent fasting), as well as different types of physical training, could also prove to be beneficial.

## 7. Conclusions

Bariatric surgeries like RYGB can effectively reduce a substantial amount of body weight, but not in a stable and metabolically healthy manner. The rapid weight loss through gastric restriction and nutrient malabsorption may result in deficiencies in plasma micronutrients (such as vitamin D, phosphorus, and folic acid), which play a role in the resolution of comorbidities and the potential for weight recovery. However, implementing a PLM that includes low- and very-low-hypocaloric diets combined with aerobic and resistance exercises from the early stages of patients’ conditioning could contribute to achieving a more sustained weight loss without exacerbating micronutrient deficiencies. As a result, morbid obesity and its associated comorbidities might be mitigated for an extended period, leading to increased patient satisfaction and longevity.

## Figures and Tables

**Figure 1 nutrients-15-03807-f001:**
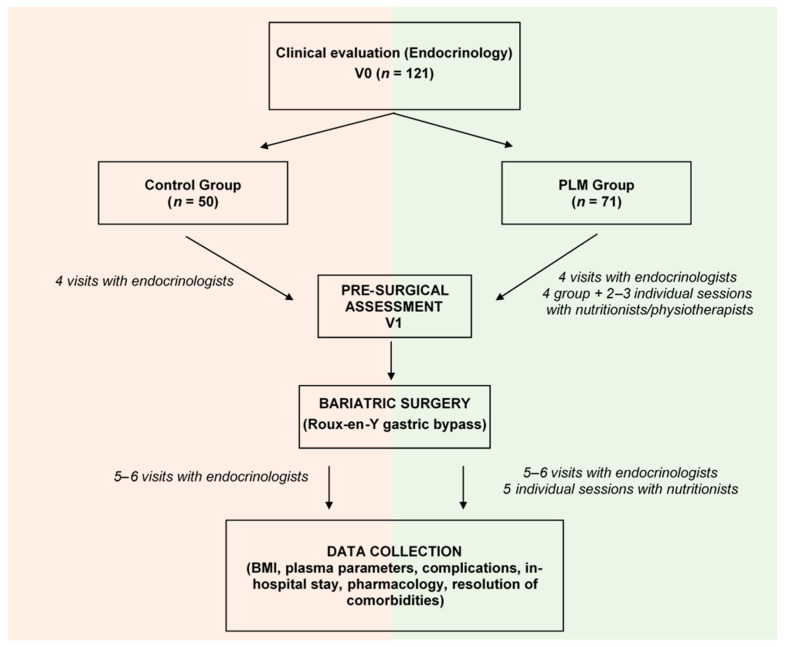
Schematic representation of the Program of Lifestyle Modification (PLM) and the conventional protocol for Roux-en-Y gastric bypass surgery. At V0, 121 obese patients were selected for the RYGB by endocrinologists. The first cohort (*n* = 50) followed a conventional procedure, while the subsequent group (*n* = 71) followed the PLM in the pre-surgical phase. Four visits (at the -12th, -8th, -4th, and -1st month before the RYGB) with endocrinologists were scheduled for control and PLM patients, and six/seven sessions (at the -11th, -10th, -9th, -8th, -6th, and -2nd month before the RYGB) with nutritionists were additionally arranged only for PLM subjects. After the RYGB, patients were also evaluated in personalized sessions with endocrinologists and/or nutritionists up to 2 years after the intervention. Five/six visits (at the 1st, 3rd, 6th, 12th, 18th, and 24th month after the RYGB) with endocrinologists were planned for control and PLM patients, and five additional sessions with nutritionists were arranged only for PLM subjects (at the 1st, 2nd, 4th, 9th, and 13th month after the RYGB). The BMI, obesity-associated comorbidities, pharmacological treatments, peri-surgical and post-surgical complications, and in-hospital stay were recorded for both procedures. Additionally, blood samples were collected for biochemical examinations.

**Figure 2 nutrients-15-03807-f002:**
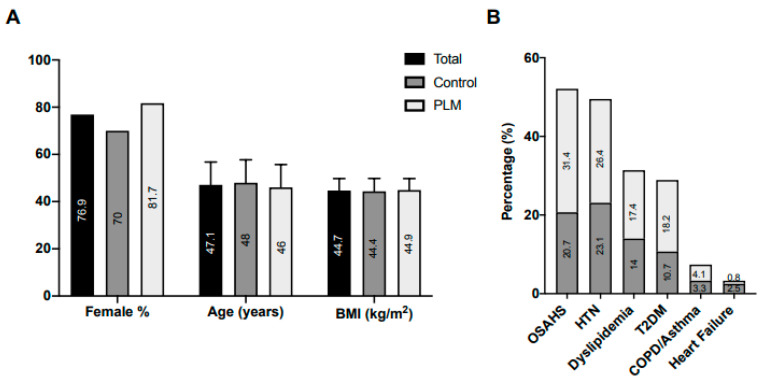
Characterization of the morbidly obese population. At V0, anthropometric parameters (**A**) and the presence of comorbidities (**B**) were recorded in obese patients assigned to either a conventional (control) protocol (*n* = 50) or the PLM (*n* = 71). No significant differences (*p* > 0.05) were found between the two groups. OSAHS stands for obstructive sleep apnoea–hypopnea syndrome; HTN represents arterial hypertension; T2DM indicates type 2 diabetes mellitus; COPD refers to chronic obstructive pulmonary disease.

**Figure 3 nutrients-15-03807-f003:**
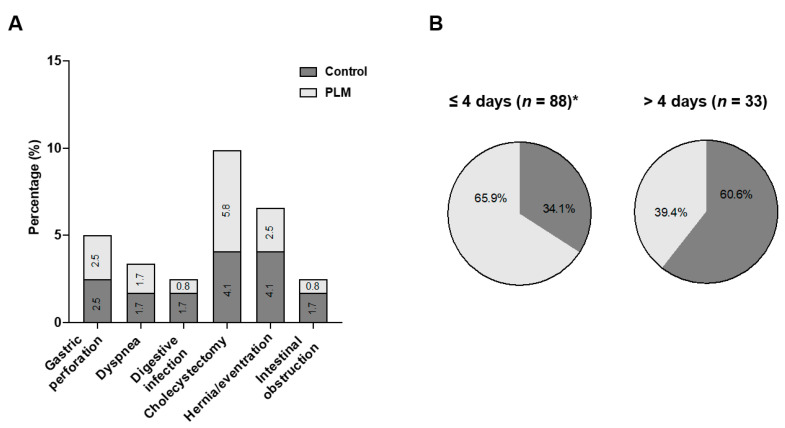
Surgical complications and in-hospital stay after the RYGB intervention. Few complications were observed during both perioperative (intra-surgery and up to 30 days post-surgery) and post-surgical (30 days to 2 years) stages after RYGB for both control and PLM subjects. No significant differences (*p* > 0.05) were detected between the two groups (**A**). The duration of the in-hospital stay (categorized as ≤4 days or >4 days) was also recorded for each group. * *p* = 0.008, PLM vs. control (**B**).

**Figure 4 nutrients-15-03807-f004:**
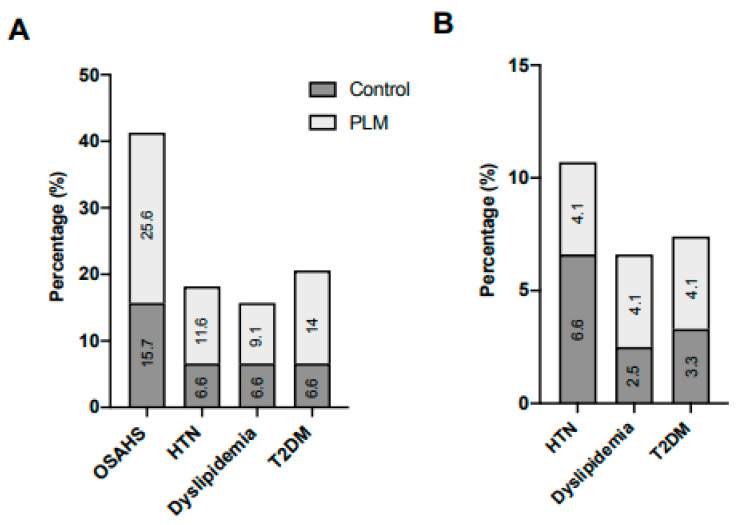
Remission of comorbidities after the RYGB. Obesity-associated comorbidities were totally (**A**) or partially (**B**) solved after the RYGB following the PLM or conventional procedure. No significant differences (*p* > 0.05) were found between both groups.

**Table 1 nutrients-15-03807-t001:** Baseline plasma markers of morbidly obese patients. Measurements of glucose and lipid profiles, ions, vitamins, folic acid, markers of liver and renal damage, and plasma proteins were taken at V0 for both control and PLM subjects. LDL and HDL represent low- and high-density lipoproteins, respectively. GOT, GPT, and PTH stand for Aspartate Aminotransferase, Alanine Aminotransferase, and parathyroid hormone, respectively. Bold, significant.

	Total (*n* = 121)	Control (*n* = 50)	PLM (*n* = 71)	*p* Value
Glucose (mg/dL)	94 (111)	95 (34)	97.5 (20)	0.533
Hemoglobin A1C (%)	5.2 (4.8)	5.95 (1.75)	6.01 (0.77)	0.147
Total cholesterol (mg/dL)	189.93 ± 36.94	191.36 ± 38.66	188.78 ± 35.96	0.714
LDL-cholesterol (mg/dL)	116.67 ± 32.97	114.2 ± 32.96	118.3 ± 33.12	0.529
HDL-cholesterol (mg/dL)	45.96 ± 11.2	46.18 ± 10.74	45.82 ± 11.51	0.901
Triglycerides (mg/dL)	154 (128)	141 (108)	113 (57)	0.261
Sodium (mmol/L)	140.63 ± 2.35	140.8 ± 2.79	140.51 ± 2.01	0.512
Potassium (mmol/L)	4.4 (0.4)	4.35 (0.45)	4.4 (0.5)	0.073
Phosphorus (mg/dL)	3.54 ± 0.6	3.72 ± 0.8	3.47 ± 0.49	0.104
Calcium (mg/dL)	9.13 ± 0.3	9.13 ± 0.35	9.13 ± 0.28	0.974
Iron (ng/dL)	75.21 ± 27.77	76.55 ± 29.28	74.29 ± 26.99	0.738
Vitamin D (ng/mL)	21.21 ± 11.25	25.24 ± 13.68	19.31 ± 9.21	**0.015**
Vitamin B12 (ng/mL)	365.56 ± 146.8	363.44 ± 148.47	400.87 ± 138.03	0.299
PTH (pg/mL)	51.23 ± 21.82	48.39 ± 24.33	54.04 ± 21.48	0.416
Folic acid (ng/mL)	6.45 ± 4.22	7.38 ± 5.47	5.74 ± 2.86	0.235
GOT (UI/L)	19 (20)	19 (10)	19.5 (9)	0.731
GPT (UI/L)	37 (19)	26.5 (20)	24 (13)	0.494
Creatinine (mg/dL)	0.94 (0.4)	0.8 (0.28)	0.7 (0.2)	**0.019**
Uric acid (mg/dL)	6.45 ± 4.22	5.66 ± 1.11	5.88 ± 1.36	0.45
Albumin (g/dL)	4.2 (0.1)	4.19 (0.3)	4.2 (0.3)	0.698
Ferritin (ng/mL)	83 (124)	97 (167.8)	75 (87.5)	0.224

**Table 2 nutrients-15-03807-t002:** Changes in BMI and essential micronutrients in obese patients at the pre-surgical stage. Both control and PLM individuals underwent pre-surgical procedures from V0 to V1 prior to the RYGB. Alterations in body weight and plasma parameters were assessed. “Dif.” denotes the differential values. Bold, significant.

V1 vs. V0	Control (*n* = 50)	PLM (*n* = 71)	*p* Value
Dif. BMI (kg/m^2^)	−0.35 (2.25)	−2.76 (2.43)	**<0.001**
%TWL	0.78 ± 4.34	6.37 ± 4.31	**<0.001**
%EWL	1.93 ± 11.64	16.33 ±10.72	**<0.001**
Dif. Phosphorus (mg/dL)	−0.35 ± 0.64	0.057 ± 0.54	**0.023**
Dif. Vitamin D (ng/mL)	0.96 (20.76)	4.3 (11.23)	0.167
Dif. Calcium (mg/dL)	−0.13 ± 0.43	0.047 ± 0.33	0.298
Dif. PTH (pg/mL)	7.57 ± 22.8	−4.78 ± 23.96	0.314
Dif. Folic acid (ng/mL)	2.45 ± 0.64	6.02 ± 11.3	0.732

**Table 3 nutrients-15-03807-t003:** Evolution of body weight after the RYGB. BMI and related parameters were compared between V4, V3, and V2 vs. V0 for both PLM and control subjects. BMI stands for Body Mass Index; %TWL represents Percentage of Total Weight Loss; %EWL denotes Percentage of Excess Weight Loss; %EBMIL signifies Percentage of Excess BMI Loss; %EEBMIL indicates Percentage of Expected Excess BMI Loss. Bold, significant.

	**Control** **(*n* = 50)**	**PLM** **(*n* = 71)**	***p* Value**
**V2 vs. V0**			
Dif. BMI (kg/m^2^)	−11.27 ± 3.21	−12.72 ± 3.88	**0.032**
%TWL	25.41 ± 6.38	28.07 ± 6.74	**0.031**
%EWL	68.32 ± 18.88	73.21 ± 16.92	0.139
	**Control** **(*n* = 50)**	**PLM** **(*n* = 71)**	***p* value**
**V3 vs. V0**			
Dif. BMI (kg/m^2^)	−13.87 ± −3.79	−15.81 ± 4.46	**0.014**
%TWL	31.3 ± 7.49	34.84 ± 7.53	**0.012**
%EWL	84.05 ± 21.76	90.65 ± 17.72	0.070
%EBMIL	94.8 ± 4.38	95.68 ± 3.71	0.138
%EEBMIL	90.3 ± 22.95	98.17 ± 19.03	**0.043**
	**Control** **(*n* = 50)**	**PLM** **(*n* = 71)**	***p* value**
**V4 vs. V0**			
Dif. BMI (kg/m^2^)	−13.53 (6.44)	−16.12 (7.5)	**0.049**
%TWL	32.72 ± 9.09	35.79 ± 8.49	0.059
%EWL	65.52 ± 17.80	69.85 ± 15.33	0.155
%EBMIL	77.60 ± 22.44	82.44 ± 18.38	0.196
%EEBMIL	94.09 ± 25.96	100.95 ± 22.07	0.120

**Table 4 nutrients-15-03807-t004:** Evolution of body weight adjusted for age, sex, differential BMI, vitamin D supplementation, and the date/season of surgery. BMI and related parameters were compared between visits and adjusted for potential confounders in both groups. BMI refers to Body Mass Index; %TWL stands for Percentage of Total Weight Loss; %EWL represents Percentage of Excess Weight Loss; %EBMIL signifies Percentage of Excess BMI Loss; %EEBMIL denotes Percentage of Expected Excess BMI Loss. Bold, significant.

	Coefficient	*p* Value	95% CI	R Squared
**V1 vs. V0**				
Dif. BMI (kg/m^2^)	−2.23	**<0.001**	−2.98 to −1.46	-
%TWL	5.68	**<0.001**	4.07 to 7.29	0.34
%EWL	14.6	**<0.001**	10.48 to 18.72	0.35
**V2 vs. V0**				
Dif. BMI (kg/m^2^)	−1.9	**0.005**	−3.2 to −0.59	0.17
%TWL	3.2	**0.008**	0.875 to 5.67	0.15
**V3 vs. V0**				
Dif. BMI (kg/m^2^)	−2.063	**0.009**	−3.59 to −0.53	0.16
%TWL	3.41	**0.015**	0.67 to 6.16	0.16
%EEBMIL	6.75	0.087	−0.99 to 14.49	0.11
**V4 vs. V0**				
Dif. BMI (kg/m^2^)	−2.21	**0.044**	−4.35 to −0.63	-

**Table 5 nutrients-15-03807-t005:** Evolution of essential micronutrients after the RYGB. Differential levels of minerals, vitamins, and associated hormones were measured after surgery across visits (V4, V3, and V2 vs. V0). Significant differences for both PLM and control subjects are presented. Bold, significant.

	**Control** **(*n* = 50)**	**PLM** **(*n* = 71)**	***p* Value**
**V2 vs. V0**			
Dif. Vitamin D (ng/mL)	6.30 ± 15.48	17.46 ± 13.53	**0.001**
Dif. Phosphorus (mg/dL)	−0.3 (0.75)	0.1 (0.7)	**0.004**
Dif. Folic Acid (ng/mL)	−0.29 (8.22)	1.52 (5.00)	**0.027**
	**Control** **(*n* = 50)**	**PLM** **(*n* = 71)**	***p* value**
**V3 vs. V0**			
Dif. Vitamin D (ng/mL)	4.93 ± 16.7	15.50 ± 14.62	**0.004**
Dif. Phosphorus (mg/dL)	0.06 ± 0.91	0.48 ± 0.62	**0.031**
Dif. Folic Acid (ng/mL)	−0.85 (6.13)	2.72 (5.73)	0.127
	**Control** **(*n* = 50)**	**PLM** **(*n* = 71)**	***p* value**
**V3 vs. V0**			
Dif. Vitamin D (ng/mL)	4.93 ± 16.7	15.50 ± 14.62	**0.004**
Dif. Phosphorus (mg/dL)	0.06 ± 0.91	0.48 ± 0.62	**0.031**
Dif. Folic Acid (ng/mL)	−0.85 (6.13)	2.72 (5.73)	0.127

**Table 6 nutrients-15-03807-t006:** Evolution of essential micronutrients adjusted for age, sex, differential BMI, vitamin D supplementation, and date/season of surgery. Plasma levels of vitamin D, phosphorus, and folic acid were compared between visits in both PLM and control patients and were adjusted for potential confounders. Bold, significant.

	Coefficient	*p* Value	95% CI	R Squared
**V2 vs. V0**				
Dif. Vitamin D (ng/mL)	8.4	**0.030**	0.85 to 15.97	0.153
Dif. Phosphorus (mg/dL)	0.56	**0.007**	0.16 to 0.96	-
Dif. Folic Acid (ng/mL)	2.9	0.242	−2.1 to 7.95	-
**V3 vs. V0**				
Dif. Vitamin D (ng/mL)	7.12	0.088	−1.1 to 15.43	0.124
Dif. Phosphorus (mg/dL)	0.39	**0.045**	−0.01 to 0.76	0.252
**V4 vs. V0**				
Dif. Vitamin D (ng/mL)	9.78	0.134	−3.07 to 22.63	-

## Data Availability

Not applicable.

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
