# Peer review of "A Program of Life-Style Modification Improved the Body Weight and Micronutrient Status in Obese Patients after Bariatric Surgery"

_nutrients, 2023, doi:10.3390/nu15173807_

Round 1

Reviewer 1 Report

Summary:

This study compares standard medical protocol with a lifestyle modification programme in a cohort of obese patients who received bariatric surgery.

The study focus is relevant due to known complications and challenges following this type of surgery, which the authors have addressed throughout.  

There are various methodological and manuscript writing issues throughout that need to be addressed prior to consideration for publication.

Comments:

Title: Consider rewording “micronutrient homeostasis” to micronutrient status” here and throughout manuscript

Abstract

-          Line 21: Add standard deviation for age and BMI

-          Line 21: Define BMI at first appearance

-          Line 22: Check grammar… Also this sentence could be removed if abstract word count is too high

-          Line 23: The term presurgical “revision” is uncommon. Amend with a clearer term

-          Line 23: Amend “increased weight loss” to “reduced bodyweight”

-          Line 24: Check grammar

-           

Introduction

-          Line 38: Check grammar

-          Line 45: Remove “associated hormones” or expand to detail relevance

-          Line 60-62: Sentence not really needed, consider removing

-          Line 65-66: Reword sentence to improve clarity

-          Line 69: Check grammar

-          More discussion is necessary describing the relevance of the utilisation of PLM in this study… This part of the introduction has been neglected and needs development

Materials and Methods

-          Line 75: How were participants allocated to each group and what was the rationale for this?

-          Line 75: Was this study registered with a clinical trial registry?

-          Line 75: What power calculations were conducted in regard to sample size and statistical analysis? There is lacking rationale for your sample size

-          Line 90: Again, suggest changing term from revision to something more common

-          Line 104: Figure 1, from the description it is not clear how often the “revisions” were? Were they completed at standardised intervals over time? Please detail. It needs to be clear for the reader the duration of time between each patient visit

-          Line 138: How was body mass measured. Please fully detail the protocol including instruments used and whether they were calibrated prior to data collection

-          Line 163: “anaerobic” exercise needs to be changed to resistance or muscle strengthening exercises based on this description.

-          Line 165: How was adherence quantified specifically? Was adherence factored into statistical analyses?

-          Line 184: Sentence unfinished

-          Table 1: There are decimal placing inconsistencies in the table, please correct

-          Tables should be placed within results section

-           

Results

-          Line 217: Add standard deviation to BMI

-          The overall word count and volume of text in the results section if very high. The authors should make efforts to make this section more concise

-          There are also too many tables, please review the journals policy on permissions for tables and figures

There are many grammatical errors throughout the manuscript. Extensive English language proof reading is required prior to consideration for publication

Author Response

Thanks for reviewing our manuscript. Here we answer point by point all required items. The changes have been included (in red) in the latest version of the work.

Sincerely,

Óscar Lorenzo

To reviewer #1

Summary:

This study compares standard medical protocol with a lifestyle modification programme in a cohort of obese patients who received bariatric surgery. The study focus is relevant due to known complications and challenges following this type of surgery, which the authors have addressed throughout. There are various methodological and manuscript writing issues throughout that need to be addressed prior to consideration for publication.

Comments:

Title: Consider rewording “micronutrient homeostasis” to micronutrient status” here and throughout manuscript

- Thanks for the suggestion. “Homeostasis” has been replaced by “status” along the work

Abstract

Line 21: Add standard deviation for age and BMI

- Ok, it has been added. Thanks

Line 21: Define BMI at first appearance

- Ok, it has been defined. Thanks

Line 22: Check grammar… Also this sentence could be removed if abstract word count is too high deleted

- Thanks, the sentence has been deleted

Line 23: The term presurgical “revision” is uncommon. Amend with a clearer term

- Ok, we agree. This term has been replaced along the manuscript.

Line 23: Amend “increased weight loss” to “reduced body weight”

      - Ok, it has been amended, as you suggest.

Line 24: Check grammar

- Sorry, the grammar and language style have been corrected along the manuscript.

Introduction

Line 38: Check grammar

- Thanks again, the grammar has been modified.

Line 45: Remove “associated hormones” or expand to detail relevance

- The sentence has been extended by including the name of related hormones.

Line 60-62: Sentence not really needed, consider removing

- Ok, we agree. The sentence has been removed, thanks.

Line 65-66: Reword sentence to improve clarity

      - Yes, sorry, the sentence has been rewritten.

Line 69: Check grammar

- Thanks, the grammar has been corrected along the manuscript.

More discussion is necessary describing the relevance of the utilisation of PLM in this study… This part of the introduction has been neglected and needs development

      - Yes, we also agree. We have described the relevance of using a PLM for bariatric surgery to improve the weight loss and micronutrient imbalance after surgery. Lifestyle modifications can incorporate changes in diet and physical activity toward a healthier feeding and performing specific aerobic and resistance exercises (Gils Contreras et al. 2020). Both strategies in combination could produce higher benefits for obese patients addressed to bariatric surgery than they can do by themselves. In addition, a nutritional follow-up might avoid metabolic imbalances related to rapid weight loss, such as reduction of fat-free mass, malnutrition, and micronutrient deficiencies (Schiavo L et al 2019). This paragraph has been added at the end of Introduction, with the corresponding references. Thanks for this annotation.

Materials and Methods

Line 75: How were participants allocated to each group and what was the rationale for this?

- Thanks again for this important issue. Our main aim was to evaluate whether the inclusion of a structured Program of Lifestyle Modification (PLM) in a bariatric process could improve the BMI at presurgical stages and later, the weight loss achieved by the intervention (i.e., RYGB). Also, we wondered whether the micronutrient status might be preserved along the peri-surgical phases by the PLM. In consequence, two cohorts of patients were evaluated; one prior to the establishment of the PLM and another posterior cohort who followed this programme. Afterwards, we analyzed all data. We have included a paragraph describing the recruitment of patients and the aim of the study in the Methodology (section: Bariatric surgery for Severe Obesity). 

Line 75: Was this study registered with a clinical trial registry?

- No, it was not.

Line 75: What power calculations were conducted in regard to sample size and statistical analysis? There is lacking rationale for your sample size

- Yes, at the end of the study we checked the statistic power by the G*Power Version 3.1.9.6 (G*Power 3: A flexible statistical power analysis program for the social, behavioral, and biomedical sciences. Behav Res Methods 2007 May;39(2):175-91. doi: 10.3758/bf03193146). Here you can see a table describing the “effect side d” and “actual power” for all significant variables. The differential BMI, VitD, phosphorous and folic acid showed a statistical power of at least 81%:

Variable

Effect Size d

Actual Power

Dif. BMI (kg/m2)

1.44

0.99

%TWL

1.29

0.99

%EWL

1.29

0.99

Dif. Phosphorous (mg/dL)

0.53

0.81

Dif. Folic Acid (ng/mL)

0.83

0.98

Vit D (ng/mL)

0.77

0.98

Also, we initially estimated the sample size for each group of patients. By considering an alpha risk= 0.05 (two-sided test), beta risk= 0.2, standard deviation= 2.5, minimum expected difference= 1.5, and a dropout rate= 0.1, the GRANMO software suggested a minimum of 49 patients for each group. Calculation of the standard deviation was based on data from a previous meta-analysis on anti-obesity interventions (Bauer K et al 2020). The estimation of the minimum expected difference was based on the desirable target weight loss for the presurgical stage for bariatric surgery, which was established in 5% (references 33 and 34 of manuscript). A minimum expected difference of 1.5 in BMI roughly corresponds to a 5% of weight loss. Thus, GRANMO software suggested at least 49 patients for each group. We included 50 subjects in the control group, and some more patients (71) in the PLM group since we believed this programme might be beneficial for these subjects. The estimation of the sample size and corresponding references have been added to the Methodology (section: Bariatric surgery for Severe Obesity). Thanks for this constructive suggestion.  

Line 90: Again, suggest changing term from revision to something more common

- Ok, “revision” has been changed along the work. Thanks

Line 104: Figure 1, from the description it is not clear how often the “revisions” were? Were they completed at standardised intervals over time? Please detail. It needs to be clear for the reader the duration of time between each patient visit

- Thanks for this important annotation. We have rewritten Figure 1 and its legend, and the corresponding part of Methodology. Indeed, all visits with endocrinologists and all sessions with nutritionist/physiotherapeutic were completed at standardized intervals over time in both groups.  

Line 138: How was body mass measured. Please fully detail the protocol including instruments used and whether they were calibrated prior to data collection

- The body weight was achieved by a mechanical column scale with eye-level beam which incorporated a telescopic rod for assessing both height and weight in one step. This precision equipment was specially designed for weighing overweight patients. Nevertheless, the equipment was calibrated prior to data collection for each patient. Then, the BMI was calculated as a measure of body mass based on height and weight applied on adult men and women (kg/m2). This description has been included in the Methodology (section: Anthropometric and biochemical parameters). Thanks again for the suggestion.

Line 163: “anaerobic” exercise needs to be changed to resistance or muscle strengthening exercises based on this description.

- Yes, we agree. It has been changed, thanks.

Line 165: How was adherence quantified specifically? Was adherence factored into statistical analyses?

- This issue was already included as a limitation of the study since we only could confirm the adherence to diet and physical exercise by directly asking to patients. Apparently, all individuals complied with the indications. However, a more reliable mechanism to ensure adherence to clinical recommendations would increase the confidence of the study.

Line 184: Sentence unfinished

- Yes, sorry, the sentence has been corrected.

Table 1: There are decimal placing inconsistencies in the table, please correct

- Yes, sorry, the table has been corrected.

Tables should be placed within results section

- Yes, figures have been placed in the right locations.

Results

Line 217: Add standard deviation to BMI

- Ok, it has been added, thanks.

The overall word count and volume of text in the results section if very high. The authors should make efforts to make this section more concise

- The Results section has been shortened to help the readers. The whole work has been revised to correct language mistakes. thanks.

There are also too many tables, please review the journals policy on permissions for tables and figures

- In principle we can include ten figures/tables, but we have asked to the journal whether we need to move to supplementary any of them for the last version. Thanks

Reviewer 2 Report

The study is interesting, although similar to many studies already done in this field. The authors report their case studies, it is already known in the literature that such paths guarantee better results for patients. 

I ask the authors to correct the following:

Line 2: I suppose the word "title:" should be removed.

Line 62: In the pdf file, the word "surgeon's" has a different character than the rest of the text. I ask the authors to check the format of the word.

Line 94: 880 UI /  Line 229: 800 UI (It is correct?)

Line 94: the international = There seems to be a double space between the two words in the pdf file.

Line 131 and Line 230: (1 g) = You add a space.

Line 138/140 and Line 244/245 and 288/289: These acronyms are already standardized in literature. Percentage Excess Weight Loss [%EWL]; Percentage Excess BMI Loss [%EBMIL]; Percentage correct-Excess BMI Loss (%C-EBMIL). Authors are asked to re-test the literature and adapt to the terms already used in the scientific field to avoid using acronyms other than those already in use and accepted by the scientific community.

Line 159: total energy uptake (TEU) (?). It is not clear, probably  = total energy intake.

Line 188/189: I advise authors to give an enter (to divide the text of the article from the text of the table).

Table 1: Table 1 seems to be a figure (even a bit blurry) in the pdf file. I recommend the authors review the table. I advise authors to write acronyms better, not like they did. Example of how best to do: TSH: Thyroid-Stimulating Hormone. I ask the authors to correct the units of measurement that report mg/dl in mg/dL and pg/ml in pg/mL.

Line 254: I advise authors to give an enter (to divide the text of the article from the text of the table).

Table 2: Table 2 seems to be a figure (even a bit blurry) in the pdf file. I recommend the authors review the table. I ask the authors to correct the units of measurement that report mg/dl in mg/dL and pg/ml in pg/mL.

Line 293/294: I advise authors to give an enter (to divide the text of the article from the text of the table).

Table 3 and Table 4: Table 3 and Table 4 seem to be a figure (even a bit blurry) in the pdf file. I recommend the authors review the table. All acronyms should be checked against those already in use in the literature.

Line 315/316: I advise authors to give an enter (to divide the text of the article from the text of the table).

Table 5: Table 5 seems to be a figure (even a bit blurry) in the pdf file. I recommend the authors review the table. 

In the text of the tables, it has sometimes been written "p value" and other times "p Value"; I ask the authors always to use the same format.

Line 340: weight loss in = Is there a double space?

Line 512: The reference has a different character from the others.

Author Response

Dear Reviewer,

Thanks for reviewing our manuscript. Here we answer point by point all required items. The changes have been included (in red) in the latest version of the work.

Sincerely,

Óscar Lorenzo

To reviewer #2

The study is interesting, although similar to many studies already done in this field. The authors report their case studies, it is already known in the literature that such paths guarantee better results for patients. 

I ask the authors to correct the following:

Line 2: I suppose the word "title:" should be removed.

- Yes, thanks, it will be removed for the last version

Line 62: In the pdf file, the word "surgeon's" has a different character than the rest of the text. I ask the authors to check the format of the word.

- Yes, it was, but we think we can even remove that sentence. Thanks again.

Line 94: 880 UI /  Line 229: 800 UI (It is correct?)

- Sorry, it was 800 UI in both cases. We have amended this mistake, thanks.

Line 94: the international = There seems to be a double space between the two words in the pdf file.

- Yes, we have corrected it

Line 131 and Line 230: (1 g) = You add a space.

- It has been corrected, thanks

Line 138/140 and Line 244/245 and 288/289: These acronyms are already standardized in literature. Percentage Excess Weight Loss [%EWL]; Percentage Excess BMI Loss [%EBMIL]; Percentage correct-Excess BMI Loss (%C-EBMIL). Authors are asked to re-test the literature and adapt to the terms already used in the scientific field to avoid using acronyms other than those already in use and accepted by the scientific community.

- Ok, we have changed the acronyms along the manuscript, figures, and legends for the accepted versions (Sabench et al. 2017 and Castro et al. 2022), as it follows: the Percentage of Total Weight Loss [%TWL: (initial weight − current weight)/(initial weight) × 100], Percentage of Excess Weight Loss [%EWL: (initial weight − current weight)/(initial weight – ideal weight) × 100], Percentage of Excess BMI Loss [%EBMIL: (initial BMI − current BMI)/(initial BMI − 25) × 100], and the Percentage of Expected Excess BMI Loss [%EEBMIL: (initial BMI − current BMI)/(initial BMI – expected BMI) × 100, being Expected BMI: (0.33 × initial BMI + 14)]. Thanks for the annotation. This has been added to the Methodology with the corresponding references.

Line 159: total energy uptake (TEU) (?). It is not clear, probably  = total energy intake.

- Yes, you are right. TEU has been substituted by TEI along the work

Line 188/189: I advise authors to give an enter (to divide the text of the article from the text of the table).

- We have separated text from legend, thanks again

Table 1: Table 1 seems to be a figure (even a bit blurry) in the pdf file. I recommend the authors review the table. I advise authors to write acronyms better, not like they did. Example of how best to do: TSH: Thyroid-Stimulating Hormone. I ask the authors to correct the units of measurement that report mg/dl in mg/dL and pg/ml in pg/mL.

- Sorry, we have revised all tables and figures, and acronyms and units have been accordantly written. The final version will include higher quality of all figures, thanks.

Line 254: I advise authors to give an enter (to divide the text of the article from the text of the table).

- Yes, we have separated text from legend, thanks again

Table 2: Table 2 seems to be a figure (even a bit blurry) in the pdf file. I recommend the authors review the table. I ask the authors to correct the units of measurement that report mg/dl in mg/dL and pg/ml in pg/mL.

- Thanks. Acronyms and units have been correctly written.

Line 293/294: I advise authors to give an enter (to divide the text of the article from the text of the table).

- Text and legend have been separated

Table 3 and Table 4 seem to be a figure (even a bit blurry) in the pdf file. I recommend the authors review the table. All acronyms should be checked against those already in use in the literature.

- Thanks. Acronyms have been corrected.

Line 315/316: I advise authors to give an enter (to divide the text of the article from the text of the table).

- Text and legend have been separated

Table 5: Table 5 seems to be a figure (even a bit blurry) in the pdf file. I recommend the authors review the table. 

- Thanks, we have revised the table

In the text of the tables, it has sometimes been written "p value" and other times "p Value"; I ask the authors always to use the same format.

- We agree, the “p value” has been amended in the figures

Line 340: weight loss in = Is there a double space?

- Ok, corrected! thanks

Line 512: The reference has a different character from the others.

- Yes, we have changed it, thanks

Round 2

Reviewer 1 Report

Authors have adequately addressed my previous review concerns 

Be sure to complete a final spelling and grammar check of the manuscript 

Author Response

Dear Reviewer,

Thanks again for reviewing our manuscript. Here we answer point by point the required items.

Sincerely,

Óscar Lorenzo

To reviewer #1

Authors have adequately addressed my previous review concerns. Be sure to complete a final spelling and grammar check of the manuscript

Thanks for the comment. We have revised the work with an English fluent researcher.

Reviewer 2 Report

I ask the authors to correct the following:

Line 2: TITLE: A program of life-style modification improved the body 2 weight and micronutrients status in obese patients after bari- 3 atric surgery = A program of life-style modification improved the body 2 weight and micronutrients status in obese patients after bari- 3 atric surgery

Line 14/32: The words "resistence" and "conclusion" have a different character than the whole text.

Table 2: Dif. Cal. md/dl = mg/dL; Dif. PTH pg/ml = pg/mL

Author Response

Dear Reviewer,

Thanks again for reviewing our manuscript. Here we answer point by point the required items.

Sincerely,

Óscar Lorenzo 

 To reviewer #2

I ask the authors to correct the following: Line 2: TITLE: A program of life-style modification improved the body 2 weight and micronutrients status in obese patients after bari- 3 atric surgery = A program of life-style modification improved the body 2 weight and micronutrients status in obese patients after bari- 3 atric surgery

Thanks for the comment. The PDF is built by the journal staff, but we will be sure tin the proof of read that the title is properly written

Line 14/32: The words "resistence" and "conclusion" have a different character than the whole text.

Table 2: Dif. Cal. md/dl = mg/dL; Dif. PTH pg/ml = pg/mL

Thanks again, we have corrected these words.